# Leveraging Artificial Intelligence in Blockchain-Based E-Health for Safer Decision Making Framework

Abdulatif Alabdulatif [1,*] , Muneerah Al Asqah [2], Tarek Moulahi [2,*] and Salah Zidi [3]

1 Department of Computer Science, College of Computer, Qassim University, Buraidah 52571, Saudi Arabia
2 Department of Information Technology, College of Computer, Qassim University, Buraidah 52571, Saudi Arabia; 411207283@qu.edu.sa
3 ISSIG, University of Gabes, Gabes 6072, Tunisia; salah_zidi@yahoo.fr
* Correspondence: ab.alabdulatif@qu.edu.sa (A.A.); t.moulahi@qu.edu.sa (T.M.)

**Abstract:** Machine learning-based (ML) systems are becoming the primary means of achieving the highest levels of productivity and effectiveness. Incorporating other advanced technologies, such as the Internet of Things (IoT), or e-Health systems, has made ML the first choice to help automate systems and predict future events. The execution environment of ML is always presenting contrasting types of threats, such as adversarial poisoning of training datasets or model parameters manipulation. Blockchain technology is known as a decentralized network of blocks that symbolizes means of protecting block content integrity and ensuring secure execution of operations.Existing studies partially incorporated Blockchain into the learning process. This paper proposes a more extensive secure way to protect the decision process of the learning model. Using smart contracts, this study executed the model's decision by the reversal engineering of the learning model's decision function from the extracted learning parameters. We deploy Support Vector Machine (SVM) and Multi-Layer Perceptron (MLP) classifiers decision functions on-chain for more comprehensive integration of Blockchain. The effectiveness of this proposed approach is measured by applying a case study of medical records. In a safe environment, SVM prediction scores were found to be higher than MLP. However, MLP had higher time efficiency.

**Keywords:** blockchain; e-health; machine learning; deep learning; smart contract; decision function





## 1. Introduction

Present-day lives require people to depend on various types of technology to assist in achieving higher levels of productivity and better operational efficiency. The continuous growth of computer-based technologies has placed them as essential pillars in new world development. Such intelligent technologies include the Internet of Things (IoT) [1] systems such as smart homes and supply chain management systems, spam filtering systems, and many others. As a critical life sector, smart healthcare systems, such as diagnostic systems or e-health decision systems, are another application of these technologies. These applications rely on Machine Learning (ML) models to detect and diagnose diseases and help disease spread prediction, such as the COVID-19 virus. ML uses models that train with heterogeneous data types to progressively develop and learn to make decisions based on their calculated outcomes. These outcomes are the decisions that can help automate routine tasks, detect abnormalities, and predict disease spreads.

Unfortunately, the outcome of this decision can encounter various threats that affect and change its value. These threats can include model parameter manipulation, poisoning attacks, and evasion attacks. The latter two are types of Adversarial Machine Learning (AML) attacks which are manipulative attacks that affect the integrity of ML datasets. A recent study of [2] applied adversarial attacks on six different COVID-19 detection systems with underlying ML Deep Neural Network (DNN) models. The authors showed that the confidence of the DNN model dropped from 91% to 9% on a subject having positive

COVID-19 results when adding random noise of black and white batches in Computed tomography (CT) scan training images. The previous experiment is one of many other examples that proved ML models' susceptibility to AML attacks.

### 1.1. Motivation and Problem Background

As shown by Figure 1, AML attacks include falsifying data samples to achieve ML model inaccuracy in classifying new data inputs [3]. A typical ML process splits the dataset between two phases, training and testing. A poisoning attack affects the training dataset, while an evasion attack injects carefully crafted samples into the testing dataset. AML research is literature designed to measure AML's impact on ML models to find a way to increase their robustness against such attacks [4]. For instance, works of [5–7] evaluated Neural Networks (NN)-based systems against different types of evasion attacks. The work of [5] performed an evasion attack on a Multi-Layer Perceptron (MLP) Intrusion Detection System (IDS), where they succeeded in dropping the model's accuracy from 99.8% to 29.87%. Moreover, Ref. [6] injected adversarial samples to fool Conventional Neural Networks (CNN) malware detection. Meanwhile, Ref. [7]'s work was successful in tricking a DNN visual recognition model into classifying adversarial inputs as benign.

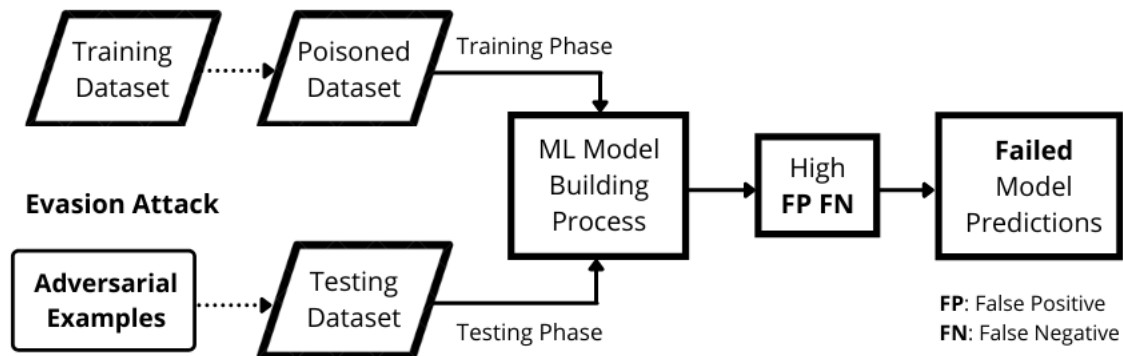

**Figure 1.** An illustration of poisoning and evasion attacks workflow.

Additionally, Ref. [8]'s work is an example of a poisoning attack that tested the susceptibility of a Support Vector Machine (SVM) spam filtering system by inserting a well-crafted label-flipped malicious sample into the training dataset.

In e-Health applications, the authors of [9] applied a poisoning attack on a LASSO regression ML model trained on a dataset that contained records of 5700 patients that predicted the dosage of Warfarin, an anticoagulant drug. Applying a 20% poisoning attack caused patients' dosages to change by an average of 139.31%. An increased dosage of Warfarin can cause severe bleeding, while a decreased dosage could cause blood clots, leading to heart attacks if the patient has a history of blood clotting [10]. This notable impact of AML attacks on people's health can severely affect other similar e-health systems [4]. A more comprehensive survey of similar studies on AML effects on other ML models, and domains can be found in [3].

The AML field of research can easily state that most types of ML are prone to adversarial attacks since it is impossible to assume that the system's environment is entirely benign. Security researchers are always on the work to deploy robust methods against AML. For example, Blockchain is an emerging technology that uses means of cryptography and decentralization to provide stable, secure, and immutable blocks of records. Multiple blocks are connected together through a hash-based procedure. This hashing procedure and the utilization of other cryptography methods have given Blockchain technology its property of protecting block content integrity [11]. More security researchers incorporate

Blockchain with ML to protect against AML. However, we believe that a more sophisticated kind of integration of the ML decision process is still needed.

### 1.2. Study Contribution and Novelty

The paper's main contributions are:

- Develop a trust-based AI framework that relies on the integration of Blockchain and ML models;
- Develop an effective method to secure the decision functions of SVM and MLP models by using immutable smart contracts.

### 1.3. Why Blockchain?

In our paper, the blockchain is used for two reasons:

- To protect the dataset against poisonous attacks. Since the used dataset is securely uploaded to the blockchain instead of publishing it in a shared repository. Indeed, the blockchain will guarantee the integrity of data;
- To protect machine learning techniques against evasion attacks. We perform this goal by embedding model decision functions as smart contracts in the Ethereum blockchain.

The rest of this paper is organized as follows: Section 2 provides a brief review of existing Blockchain-adopted ML research. Moreover, it explains the necessary background details of Blockchain and ML models. The applied system flow is illustrated in Section 3, while Section 4 elaborates on the implementation steps of the proposed system, including the reconstruction of SVM and MLP decision functions. Lastly, Section 5 analyzes and discusses the performance of the proposed system.

## 2. Background Study

The revolutionized growth in Blockchain technology has encouraged its emergence with ML solutions. This section outlines related works and provides a brief technological preview.

### 2.1. Related Literature

Several studies cover combining Blockchain with ML to solve security and privacy issues in the literature. We summarize these integrations into three main categories, NN-based integration, partial integration of Blockchain, and Blockchain with Federated Learning. One of the NN-based integration examples is called DeepRing, where authors of [12] designed each NN layer to be presented as a Blockchain block to protect against tampering attacks. Although this integration stood robust against a tampering attack that downgraded a regular CNN performance by 20.71%, its application is limited to NN-based ML models. Other research included the partially separated integration of ML with Blockchain [13]. One example is [14]'s work of combining ML with Blockchain for a more efficient and safer COVID-19 vaccine supply chain. Their solution queried records from the Blockchain to feed them into a separate Long Short Term Memory (LSTM) classifier. The demand forecasting LSTM helped preserve 4% of the vaccine ratio, 6 million vaccine doses.

Most Blockchain-integrated ML solutions focus on employing the technology to protect the privacy of the ML model. Studies with such scope deployed Federated Learning (FL) [15], or as can be known as Decentralized ML (DML) [16], where a centralized server collects and aggregates learning parameters among participating nodes.

The majority of found FL-based integration, such as works of [17–19], relied on off-chain execution of ML training while applying different types of consensus algorithms to manage work among nodes. This partial application of the ML decision process is due to the metered usage and storage of Blockchain, which can result in the prosperous implementation of the whole ML fitting and decision process.

One other noticeable example is the collaborative deep learning framework called DeepChain. Similar to the previously mentioned studies, Ref. [20] designed DeepChain for nodes to train a global model locally and then upload their gradients through a smart contract. They also relied on consensus algorithms for control updating the model's gradients which are averaged and broadcasted again for the next learning iteration. Table 1 summarises existing related work and their limitations.

**Table 1.** Summary of existing solutions incorporating ML with Blockchain and their limitations.

| Ref. | Summary | Limitation |
|------|---------|------------|
| [14] | Combined Blockchain and ML to design a more efficient COVID-19 supply chain system. | Separate integration of ML modules. No focus on security/privacy |
| [12] | Designed DeepRing, where each NN layer is presented as a Blockchain block to protect against tampering attacks. | Not suitable for other none NN-based ML models |
| [21] | Developed dynamic malware detection system based on behavioral logs using Deep learning and Blockchain | No consideration to poisoning attacks prevention. No privacy consideration |
| [22] | Used DanKu protocol to build a malware detection system on Blockchain | Resource wastage due to using DanKu protocol. No consideration to poisoning attacks |
| [18] | Used FL with Blockchain to mitigate end-point corruption attacks. | The consensus committee is only one member who is considered non-hostile |

As shown in Table 1, the studies incorporating ML with Blockchain are scarce. At the same time, most of these integrations focused on collaborative learning and partially included Blockchain in the ML decision process. None of the found literature included the decision process to be performed on-chain. To our knowledge, there is still no ML decision integration with Blockchain. This study proposes the further inclusion of the decision function in smart contracts to achieve a more reliable and secure decision process.

### 2.2. Technological Background

Since the apparition of bitcoin as an electronic cash system in 2008, researchers and developers have been working to enhance Blockchain systems to be the main pillars of future systems [23].

### 2.2.1. Blockchain Technology

The term "Blockchain" started to be popularly used to refer to the technology presented in Nakamoto's paper. Although it was considered a breakthrough in technology, concepts of Blockchain, such as cryptography and hashing, were explored way before Bitcoin's publication [24].

Blockchain has many definitions, but it can be defined as the technology that uses block-type data structure to store data, uses consensus algorithms to generate and update the distributed ledger, and uses encryption to ensure security during transmission [25]. To be put in other words, Blockchain is a Peer to Peer (P2P) network, where nodes share a distributed ledger [26].

Blockchain protects the integrity of the ledger content through hashing; each block's hash connects with the previous block, and a small change in any block's content will not go unnoticed [11]. Consensus algorithms are rules and agreements which the P2P network uses to draw verdicts on the new block's validity to the ledger. Nodes calculate a cryptography challenge, called a nonce, to prove the block's validity. Once the nonce is validated, the new block is added. The previous operation is also called mining. Different consensus algorithms rely on various intensives to encourage mining nodes [26].

### 2.2.2. Smart Contracts

Smart contracts control the ledger's state by dynamically sending specific transactions through execution conditions and logic. When conditions are met, the execution logic is invoked [25]. Smart contracts enabled Blockchain to manipulate digital assets inside the Blockchain. It was first implemented in the Ethereum Blockchain platform in 2015 [26]. Smart contracts can be written in Solidity or Vyper programming languages, and they execute on the Ethereum Virtual Machine (EVM). Smart contracts enabled the development of distributed applications (dApp) [26].

### 3. Proposed Methodology

This section presents the proposed model followed by experiments approve to achieve ML execution security. Figure 2 shows the main steps to achieve the objective of this study. A smart contract writes the dataset to Blockchain. There are two ways to store the dataset. One way is to store the dataset off-chain while keeping the hash value of the dataset on-chain. In contrast, the whole dataset can be stored on-chain. The choice of on- or off-chain storage depends on the size of the dataset; storing a vast dataset on-chain can be costly. Direct complete on-chain execution of the ML fitting process was not applicable due to the following reasons:

1. EVM is still immature to execute complex ML operations;
2. Storing the large-size of ML libraries is considered costly in terms of deployment and runtime.

The proposed methodology suggests that the ML fitting process be executed off-chain in a local client. Fitted models' parameters will then be extracted to be stored on-chain.

Since the dataset followed a classification problem, this study employed two classifiers, SVM and MLP. SVM is a popular ML classifier that uses influence functions to find a hyperplane that best separates two data classes. On the other hand, MLP is a deep learning model that is a type of feedforward NN that uses a set of nodes organized into multiple layers to draw prediction conclusions.

A smart contract writes the model's parameters to Blockchain. These written parameters are used to reversely construct the decision function of the ML model on-chain which will help to classify a new given datapoint vector.

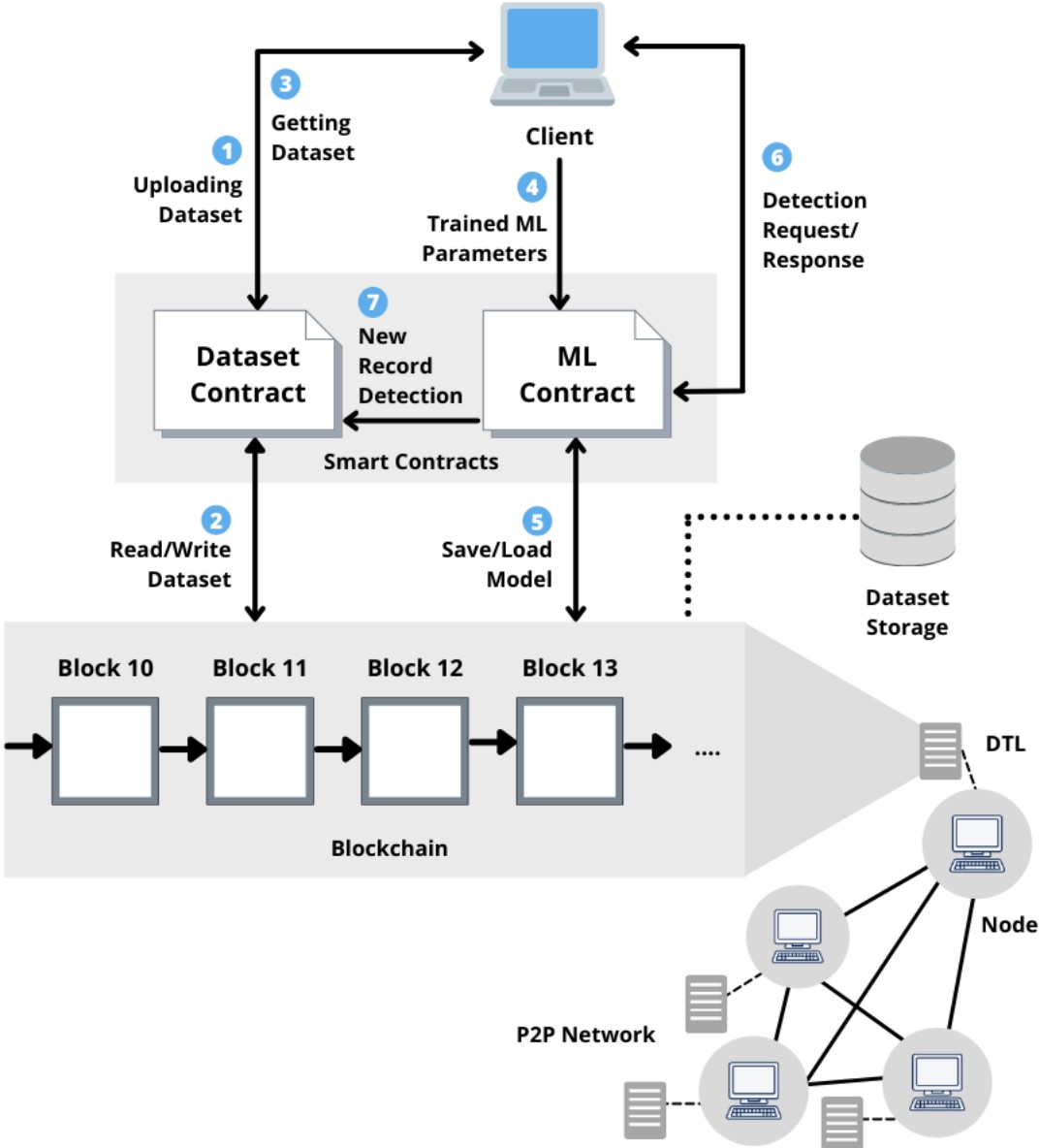

**Figure 2.** An overview of the proposed model along with entities interaction and workflow.

## 4. Implementation

This section elaborates on the implementation details of the proposed methodology. The implementation device was an AMD64 3.20 GHz CPU with a 16 GB RAM computer, and the implementation of Blockchain was performed using Ganache [27].

### 4.1. Experimental Dataset Setup

The dataset used in this experiment is the Pima Indians Diabetes dataset, which contained the measurements of labeled 21 or older 768 females, where 268 records were of females with type 2 diabetes, and the rest were healthy females.

In preparation for ML fitting, two copies of the dataset were prepared. Copy 1 of the dataset was set to be preserved on-chain. On the other hand, an evasion attack of manual label-flipping was implemented on copy 2 of the dataset, where healthy records appeared with diabetes labels and vice versa. Almost 33% of records were manipulated and labeled as poisonous samples, and the rest were labeled as normal records.

### 4.2. Writing Dataset

In this implementation, the data storage was on-chain storage since the 24 kilobytes size of the experiment dataset was quite manageable. Algorithm 1 shows the pseudo code of writing copy 1 of the dataset procedure, where each record of copy 1 of the dataset was converted to Javascript Object Notation (JSON) string format before writing them to the ledger. The storing of dataset on Blockchain will help to protect its integrity against poisonous attacks. In addition, it makes it available for use with a high safety level.

---

**Algorithm 1** Write dataset to the distributed ledger

---

     **Input:** *file* dataset file
1:  **procedure** UPLOADDATASET
2:     **for** $i \leftarrow 1$ to length of *file* **do**
3:         $Record \leftarrow file[i]$ in JSON
4:         DatasetProtect.writeRecord(*Record*)

---

Line 4 in Algorithm 1 is a call to a setter function in the *DatasetHandler* smart contract.

### 4.3. Reading Dataset

Loading records from the ledger is less expensive and more straightforward. Algorithm 2 shows the steps of the loading procedure.

---

**Algorithm 2** Read dataset from the distributed ledger

---

     **Output:** *recordList*[] list of records in JSON
1:  **procedure** LOADDATASET
2:     $latestID \leftarrow$ DatasetHandler.dCount
3:     **Init** *recordList* as Array
4:     **for** $i \leftarrow 1$ to *latestID* **do**
5:         $R \leftarrow$ DatasetHandler.readRecord($i$)
6:         **Push** $R$ to *recordList*[]
7:     **return** *recordList*[]

---

Line 2 in Algorithm 2 is a call to a smart contract getter function to obtain the latest record ID value. Line 5 is a call to a getter function inside the smart contract to retrieve the record JSON string. This procedure returns a list of JSON strings representing all the dataset records which then can be converted to any other format for ML model training, a Comma-Seperated Value (CSV) file for instance.

### 4.4. Parameters Extraction and Preservation

At this stage, an off-chain client loads the dataset copy 2 and starts the ML fitting process to detect poisonous records. In this use case of application, an additional standardizer is implemented to normalize the dataset.

#### 4.4.1. Scalar Parameters

Copy 1 of the dataset is scaled using a standard scalar that uses means and standard deviation values to standardize data to values close to 1 and –1 . The extraction of the fitted scaler produced two vectors with a length of 9, which is the same number as the dataset's features. The two vectors are the *means* for the means values, and the *vars* for the variances. These vectors were set and preserved on-chain for future use.

#### 4.4.2. SVM Parameters

SVM was trained using the radial basis kernel function (RBF), which calculates the Euclidean distance between vectors. After the training process is complete, the following model's training parameters were extracted to be preserved on-chain as shown in Table 2.

**Table 2.** SVM extracted parameters.

| Name | Description | Data Type |
|---|---|---|
| support_vectors_ | Datapoints defining hyperplane decision boundaries placements | $9 \times 395$ Decimal Matrix |
| _dual_coef | Weights of support vectors | $1 \times 395$ Decimal Array |
| _intercept | The bias | Decimal |
| _gamma | To handle non-linear classification | Decimal |

Support vectors were saved individually through a loop similar to the one previously used to store dataset records due to the HyperText Transfer Protocol (HTTP) limitations to pass the sizeable multi-dimensional list to the chain in one transaction.

### 4.4.3. MLP Parameters

MLP training process included two hidden layers of sizes 5 and 2 over 1000 epochs. The feed-forward deep learning NN relied on a nonlinear activation function, known as the Rectifier Linear Unit activation function (reLU). Likewise, Table 3 shows MLP parameters stored on-chain after the training is complete.

**Table 3.** MLP extracted parameters.

| Name | Description | Data Type |
|---|---|---|
| coefs_ | Weights of neuron's inputs in three layers, two input layers, and the output layer | $5 \times 9$, $2 \times 5$, $1 \times 2$ Decimal Matrices |
| intercepts_ | Biases of each neuron in three layers | $1 \times 5$, $1 \times 2$, $1 \times 1$ Decimal Arrays |

The learning settings of both SVM and MLP were found to be best in this case of classifying poisonous diabetes data records by balancing performance and avoiding overfitting.

### 4.5. ML Detection Implementation

This study proposes preserving efficiency by manually deploying decision functions built from both algorithms' previously-stored parameters to carry the detection of new data input on-chain. As mentioned earlier, the entire algorithm learning process on-chain was not cost-efficient.

### 4.5.1. Scalar Standardization Function

A new data point needs to be scaled first to be classified by the ML model. The scalar applies the standardization function below to the means and variances values acquired in the previous step:

$$z = \frac{x - \mu}{\sigma} \tag{1}$$

where $z$ is the scaled vector, $x$ is the input vector, $\mu$ is the mean value, and $\sigma$ is the standard deviation value, which is the square root of the variance value. Algorithm 3 shows the scaling procedure steps.

---

**Algorithm 3** Standardize new data point vector.

---

    **Input:**
*vector* integer input vector,
*means* means array,
*vars* variances array
    **Output:**
*scaledVector* scaled decimal output vector

  1:  **procedure** SCALEDATA
  2:     $stds \leftarrow$ sqrt(*vars*)                         ▷ call math sqrt() function
  3:     **for** $i \leftarrow 1$ to length of *vector* **do**
  4:         $vector[i] \leftarrow vector[i]$ in Decimal     ▷ convert integer to decimal format
  5:         $scaledVector[i] \leftarrow (vector[i] - means[i])/stds[i]$

---

It was not possible to pass decimal arrays directly to smart contract functions during the implementation of the proposed methodology. For this reason, as seen in line 4 in Algorithm 3, every decimal array was passed as an integer and then converted back to decimal on-chain for further calculations.

4.5.2. SVM Decision Function

An RBF-kernel SVM's decision function returns values close to (–1,1) and is generally described in the math equation below:

$$h^* = (\mathbf{x}^\star \phi(x)) + w_0^\star$$
$$h^* = \sum_{i \in P_S} a_i^\star u_i \cdot K(\mathbf{x}_i - \mathbf{x}) + w_0^\star \tag{2}$$

In the above representation, $h^*$ is the decision function, $a_i^\star$ is the value of the coefficients, $u_i$ is the support vector output of the kernel function $K$, $x$ is the new data point vector, $\mathbf{x}_i$ is the support vector, and $w_0^\star$.

Since the kernel function of this SVM implementation was the RBF function, it has the mathematical representation as follows:

$$K(\mathbf{x}, \mathbf{x}') = \mathbf{exp}(-\gamma \|\mathbf{x} - \mathbf{x}'\|^2) \tag{3}$$

In the above equation, $\gamma$ is the gamma value acquired previously. The RBF kernel function returns the product of negative gamma with the Frobenius norm of two input vectors. Function **exp**() is the exponent of Euler number, $e$.

$$\|\mathbf{x}, \mathbf{x}'\|_F = \sqrt{\sum_{i=1}^{m} \sum_{j=1}^{n} |a_{i,j}|^2} \tag{4}$$

The above math representation shows that the Frobenius norm, $F$, is the square root of the summation of two input vectors, $x, x'$, squared difference, $a$.

Smart contracts did not provide complex decimal math libraries support for exp() function, nor did they allow execution for decimal numbers to be the base or the exponent of exponentials. This study came with the workaround to use the Taylor Maclaurin series to calculate the exponential of decimals:

$$f(x) = \sum_{n=0}^{\infty} \frac{x^n}{n!} \tag{5}$$

The above math representation shows that the Taylor Maclaurin series is the summation of a number $x$ raised to power $n$ divided by the factorial of that $n$.

Taylor Maclaurin series is an approximation calculation, which means that calculating $e^x$ using the summation of an infinite number $n$ iterations will progressively produce a value closer to the actual value.

The following Algorithm 4 shows the steps of calculating the Taylor Maclaurin series in fifty rounds of calculations, as it was better suited for execution efficiency while obtaining more precise values. Algorithm 5 shows the steps of calculating the Frobenius norm of Equation (4).

---

**Algorithm 4** Approximate $e^x$ using Taylor Maclaurin series

> **Input:** $x$ decimal number
> **Output:** *result* decimal number

1: **procedure** TAYLOR
2:    $term \leftarrow result \leftarrow n \leftarrow 1$
3:    **for** $i \leftarrow 1$ in **range(50) do**
4:       $term \leftarrow (term * x)/n$
5:       $n \leftarrow n + 1$
6:       $result \leftarrow result + term$
7:    **return** *result*

---

**Algorithm 5** Frobenius norm of two vectors

> **Input:**
> $x$ decimal support vector,
> $z$ decimal input vector
> **Output:**
> *sum* decimal number

1: **procedure** FNORM
2:    $sum \leftarrow 0$
3:    **for** $i \leftarrow 1$ to length of $x$ **do**
4:       $y \leftarrow (x[i] - z[i])^2$
5:       $sum \leftarrow y + sum$
6:    **return** sqrt($sum$)

---

Equation (3) of the kernel function is calculated by using the Frobenius norm and the Taylor Maclaurin series, as shown by Algorithm 6.

---

**Algorithm 6** RBF kernel of support vector and input vector

> **Input:**
> $x$ decimal support vector,
> $z$ decimal input vector,
> $g$ gamma value
> **Output:**
> $y$ decimal number

1: **procedure** RBF
2:    $norm \leftarrow \text{Fnorm}(x,z)^2$            ▷ call Fnorm procedure
3:    $y \leftarrow (g * norm) * (- - 1)$
4:    **return** Taylor($y$)            ▷ call Taylor procedure

---

It is worth noting that the square root step in Algorithm 5 line 7 cancels the squaring step in Algorithm 6, line 2. For this reason, these steps were omitted in the smart contract code implementation.

Algorithm 7 applies Equation (2), where it takes an integer vector input and returns a value > 0 or <0. If the output of this function is larger than zero, a positive number, it has a label of class 1. Alternatively, if the output is less than zero, a negative number, it has a label of class 0.

---

**Algorithm 7** SVM decision function

---

**Input:**
$x$ integer input vector,
$sv$ support vectors matrix,
$df$ dual coefficients array,
$incpt$ intercept decimal value,
$g$ gamma decimal value
**Output:**
$v$ decimal number

1: **procedure** SVMDECFUN
2:     $z \leftarrow \text{ScaledData}(x)$                                          ▷ call ScaleData procedure
3:     **init** $rbfList[]$ as Array
4:     **for** $i \leftarrow 1$ to length of $df$ **do**
5:         $rbf \leftarrow \text{RBF}(sv[i], z, g)$                               ▷ call RBF procedure
6:         **push** $rbf$ to $rbfList[]$
7:     $sum \leftarrow 0$
8:     **for** $i \leftarrow 1$ to length of $df$ **do**
9:         $y \leftarrow df[i] * rbfList[i]$
10:         $sum \leftarrow sum + y$
11:     $v \leftarrow (sum + incpt) * (- - 1)$
12:     **return** Taylor($v$)

---

### 4.5.3. MLP Decision Function

MLP has an input layer of 9-dimensions, and two hidden layers of 5 and 2 dimensions, and since it is solving a binary classification problem, it has a 1-dimension output layer.

The decision function of MLP concludes a series of addition and multiplication to classify an input. In this calculation, each hidden neuron's value equals the linear summation of all previous layer's neurons' values multiplied by their coefficients, or the weights between the neuron's layer and the last layer. An additional value of intercept, or bias, is added to this summation:

$$h_i^{(1)} = \phi(\sum_j x_j w_{i,j} + b_i^{(1)})$$

$$h_i^{(2)} = \phi(\sum_j h_j^{(1)} w_{i,j} + b_i^{(2)}) \tag{6}$$

$$y_i = \phi(\sum_j h_j^{(2)} w_{i,j} + b_i^{(3)})$$

Equation (6) calculates MLP decision function where $h_i^n$ is the neuron $i$ value in the $n$th layer. This implementation includes two layers and a final output layer with one neuron, $y_i$, which gives the final summation value. $\phi()$ is the nonlinear activation function that calculates the neuron's value by a weighted sum, where $x_j$ is the input features vector, $h_j^n$ is the neurons' values at layer $i - 1$, $w_{i,j}$ is the weight, and $b_i^n$ is the intercept of neuron $i$ at the $n$th layer.

$$f(x) = max(0, x) \tag{7}$$

This MLP implementation follows a Rectifier Linear Unit activation function (reLu), which, as illustrated by Equation (7), returns the max between 0 and the weighted sum, $x$, of a neuron.

Algorithm 8 shows the steps in calculating the Equation (6) decision function of this paper's implementation of MLP by using a three-level loop. It is worth noting that the current development of smart contracts did not allow for multi-loop applications. For this reason, the inner loops were applied and called separate functions inside the contract.

---

**Algorithm 8** MLP decision function

---

**Input:**
*x* integer input vector,
*w* weight matrices,
*b* biases array
**Output:**
*v* decimal number

1: **procedure** MLPDecFun
2:     $z \leftarrow$ ScaleData($x$)                                  ▷ ScaleData procedure
3:     **init** $A[]$ as Array
4:     **init** $B[]$ as Array
5:     **for** $i \leftarrow 0$ to $i \leftarrow 2$ **do**
6:         **if** $i \leftarrow 1$ **then**
7:             **for** $j \leftarrow 0$ to $j \leftarrow 4$ **do**
8:                 $xSum \leftarrow 0$
9:                 **for** $k \leftarrow 0$ to $k \leftarrow 8$ **do**
10:                     $xw \leftarrow z[k] * w[i][k][j]$
11:                     $xSum \leftarrow xSum + xw$
12:                 $a \leftarrow xSum + b[i][j]$
13:                 $a \leftarrow \max(0,a)$
14:                 **push** $a$ to $A[]$
15:         **if** $i \leftarrow 1$ **then**
16:             **for** $j \leftarrow 0$ to $j \leftarrow 1$ **do**
17:                 $aSum \leftarrow 0$
18:                 **for** $k \leftarrow 0$ to $k \leftarrow 4$ **do**
19:                     $aw \leftarrow A[k] * w[i][k][j]$
20:                     $aSum \leftarrow aSum + aw$
21:                 $b \leftarrow aSum + b[i][j]$
22:                 $b \leftarrow \max(0,b)$
23:                 **push** $b$ to $B[]$
24:         **if** $i \leftarrow 2$ **then**
25:             $bSum \leftarrow 0$
26:             **for** $k \leftarrow 0$ to $k \leftarrow 1$ **do**
27:                 $bw \leftarrow B[k] * w[i][k]$
28:                 $bSum \leftarrow bSum + bw$
29:             $y \leftarrow bSum + b[i]$
30:     $y \leftarrow \max(0, y)$                                  ▷ last reLU function
31:     **return** $y$

---

Eventually, the MLP decision function returns a decimal value of 0 or >0. If the output is greater than zero, the classification label is 1; otherwise, it is 0.

The number of iterations in each loop is related to the number of neurons in each layer of MLP, and Algorithm 8 was tailored according to this paper's MLP implementation; a different implementation should follow different specifications accordingly.

## 5. Results and Discussion

The evaluation and analysis of the proposed system's execution steps are divided by analyzing the experimental results and the decision function execution measurements.

After applying the proposed methodology, several results were obtained to measure the proposed system's efficiency.

### 5.1. ML Detection Performance

As previously mentioned, this study applied two classification models to develop a poisonous record detection system. Table 4 shows SVM and MLP performance measurement metrics.

**Table 4.** ML Classifiers Performance.

| Classifier | Accuracy | Precision | Recall | F1-Score |
|------------|----------|-----------|--------|----------|
| SVM | **0.81** | 0.86 | 0.72 | 0.74 |
| MLP | 0.71 | 0.67 | 0.61 | 0.62 |

Figure 3 shows the Receiver Operating Characteristics (ROC) curve, which shows the relation between the true positive rates (TPR) and the True Negative Rates (TNR) of the two classifiers. The two classifiers' performances show that SVM achieved higher accuracy scores of 81% compared to MLP's 71% accuracy score. SVM also obtained better TPR scores than those MLP.

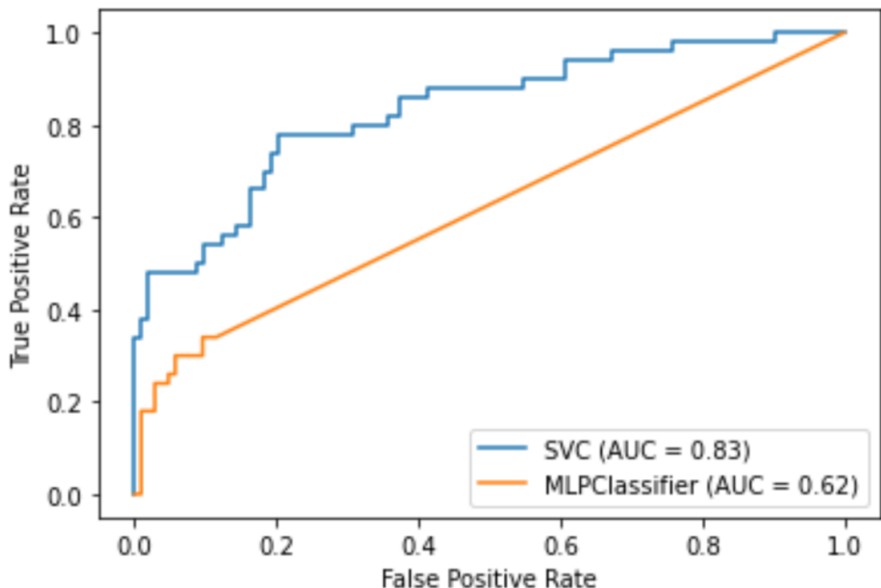

**Figure 3.** ROC curve for SVM and MLP classifiers.

*5.2. Smart Contract Performance*

This paper employs Blockchain technology through smart contracts. This subsection measures the performance and cost of such employment.

Table 5 shows details of the deployed smart contracts, including the deployment cost in gas. ML models of SVM and MLP were separately implemented to evaluate their performance better. Table 4 shows that the SVM contract has the most expensive deployment.

**Table 5.** Smart Contract Deployment.

| Smart Contract | Language | Gas Cost (gwei) |
|----------------|----------|-----------------|
| Dataset Handler | Solidity | 357,364 |
| SVM Model Handler | Vyper | 3,320,656 |
| MLP Model Handler | Vyper | 1,046,866 |

There was no clear way to measure the performance of smart contract functions regarding the CPU performance of the EVM. For this case, this study chooses to follow the judgment of each procedure's performance based on the elapsed time taken to complete each operation.

Table 6 shows the elapsed time for each read and write operation in the proposed methodology. It is worth noting that these measurements were taken on a local Blockchain network, with each transaction mined instantly to avoid any additional time latency.

**Table 6.** Elapsed time for methods' executions.

| Procedure | Method | Run Time (s) | Expected (s) | | Time Avg (s) |
|---|---|---|---|---|---|
| | | | **Runs** | **Time** | |
| Upload Dataset | writeRecord | 0.57844 | 768 | 444.24 [1] | 934.95 [2] |
| | uploadDataset | 1425.66631 | - | | |
| Load Dataset | readRecord | 0.16492 | 768 | 129.6246 [1] | 114.41 [2] |
| | loadDataset | 99.1903 | - | | |
| Write Scalar Parameters | setScalar | 20.05179 | | - | 20.05 |
| Write SVM Parameters | setSupportVector | 1.68084 | 395 | 663.9318 [1] | 634.5 [2] |
| | setSupportVectors | 605.0704 | - | | |
| | setSVM | 0.47049 | - | | 0.47049 |
| | setDualCoef | 46.240966 | - | | 46.240966 |
| | | | | | Total = 681.21 |
| Write MLP Parameters | setFirstWeights | 40.60692 | | - | 40.60692 |
| | setSecondWeights | 11.84798 | | - | 11.84798 |
| | setThirdWeights | 2.745 | | - | 2.745 |
| | setBiases | 7.25123 | | - | 7.25123 |
| | | | | | Total = 62.45 |

[1] Calculated expected elapsed time by single run time × number of items' runs. [2] Averaged time between calculated expected and actual time of operation execution.

The elapsed time analysis showed that the write operation with smart contract setter functions has low time efficiency than the getter functions. The longest time was to set SVM's support vectors since writing a single support vector takes about two seconds to complete.

As mentioned before, the smart contract did not allow for the direct passing of decimal vectors; vectors were sent as integers and then converted back to decimals. These additional conversion steps could be the reason for the setting support vector procedure's low time efficiency; a more thorough CPU analysis could determine the cause for such latency.

### 5.3. Decision Function Performance

As mentioned before, this study chose to lower the implementation cost and only deploy the ML model's decision functions (DFs). Table 7 shows SVM and MLP performance details, including the execution cost and the elapsed time with average of 10 runs.

**Table 7.** DF performance details.

| DF | Execution Cost (gwei) | Elapsed Avg (s) |
|---|---|---|
| SVM DF | 16,495,436 | 8.4415 |
| MLP DF | 316,215 | 0.1914 |

A similar script was applied to time the completion of each deployed classifier prediction to calculate the smart contract DF. Figure 4 shows the elapsed time in seconds of both SVM and MLP in comparison with the built-in functions executed on the test machine.

MLP performed better than SVM regarding deployed DF time-efficiency and cost-efficiency. While obtaining a classification with the SVM's smart contract DF takes almost 10 s, it takes less than a second to obtain a classification with MLP. However, both classifiers' DFs fell behind in comparison with the client test machine's performance, which could be because of the humble EVM abilities to execute complex math methods in contrast with the test machine's abilities.

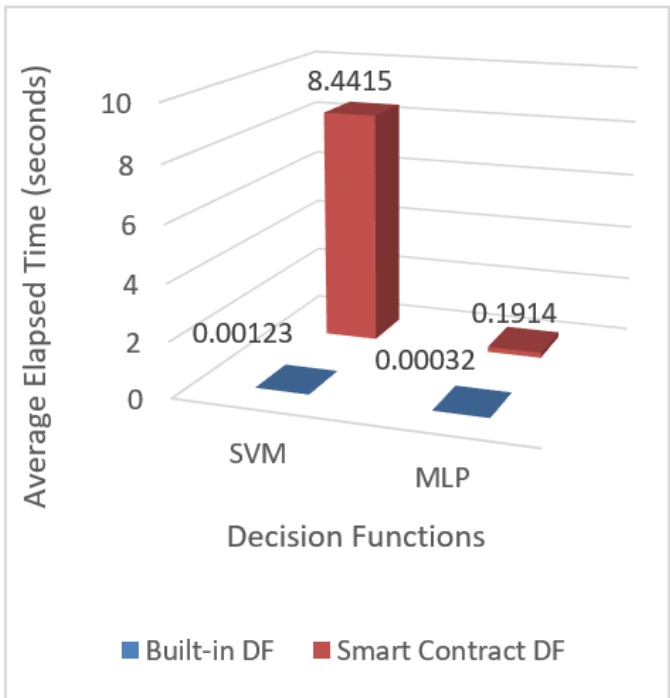

**Figure 4.** Elapsed time of SVM and MLP DFs.

*5.4. Overall Performance Comparison*

This section discusses the proposed methodology's overall performance by examining three essential perspectives, security, cost efficiency, and run-time efficiency.

- Security Perspective: No system can be 100% secured, and smart contracts are not an exception to that rule. Smart contracts can contain numerous security vulnerabilities: re-entrancy, unhandled exceptions, Integer Overflow, and unrestricted action [28]. Many reports and literature have discussed and studied such vulnerabilities. For instance, a study of [29] that evaluated 1.2 million smart contracts found that only 4% of real-world smart contracts are affected by the integer bug vulnerability.
  Authors of [28] argued that, even if the smart contract contained reported vulnerabilities, it does not mean it can be exploited in practice. They analyzed 23,327 vulnerable contracts and found that only 1.98% are exploited. By remarking on this, the proposed smart contract-based system can preserve the safety of execution by 96–98%.
- Cost Efficiency Perspective: The proposed methodology of integrating ML into Blockchain by only deploying the decision function of the ML model preserves the cost of storing large ML libraries and the execution cost of training the ML model entirely on-chain. Training the ML model off-chain and only deploying the decision process on-chain enhances the ML-integrated Blockchain cost efficiency.
- Run-time Efficiency Perspective: The run-time of the proposed system depends on EVM execution abilities. EVM is a run-time virtual machine with limited resources, which causes it to take longer to execute basic programming procedures, such as loops. Although the proposed system can preserve higher security and better cost efficiency, it has low run-time efficiency, which is believed to be increased with EVM development.

## 6. Conclusions

The proposed methodology in this paper provides a more exhaustive and efficient way to integrate AI abilities with Blockchain. In fact, it is more important to secure the process of using ML than improve ML itself. Blockchain provides the means of hashing to ensure this process of ML model decision integrity. This study proposed the flow of preserving trained models' gradients on-chain and reverse-engineering the decision function of SVM and MLP models on-chain.

During implementation, SVM proved to be more accurate than MLP in poisonous records detection. However, MLP achieved a higher time efficiency. Joining Blockchain to ML is very useful in sensitive domains like e-Health, which affects human life. This joining provides a safe environment for ML techniques for decision-making.

For future work, a more improved ML training procedure will increase the detection performance of the system. In addition, a layer of consensus could be added to control the uploading of training parameters. The obstacles faced by this study were primarily because of the immature EVM execution of complex math. When EVM becomes more efficient, the application of the proposed methodology will achieve better performance. Additionally, this integration of smart decisions gives Blockchain and smart contracts the AI ability to intelligently classify and detect, which is applicable to various Blockchain scenarios besides poisoning attack detection.

In our framework, the role of blockchain is to secure the decision process by using smart contracts. As for the consensus mechanisms among the connected nodes, it can be future work to further strengthen the learning process.

Finally, this study presented a prototype for the future incorporation of ML with Blockchain to take both technologies further in their evolution by securing the decision process.

**Author Contributions:** M.A.A., T.M. and S.Z. contributed to the conceptualization, methodology and writing—original draft. T.M. and A.A. contributed to methodology, project administration, visualization, and writing—review and editing. All authors have read and agreed to the published version of the manuscript.

**Funding:** This research was funded by Deputyship for Research and Innovation, Ministry of Education, Saudi Arabia Grant No. QU-IF-04-01-28436.

**Data Availability Statement:** The dataset used to support the findings of this study has been deposited in the website of kaggle repository (https://www.kaggle.com/uciml/pima-indians-diabetes-database, accessed on 8 January 2023).

**Acknowledgments:** The authors extend their appreciation to the Deputyship for Research and Innovation, Ministry of Education, Saudi Arabia for funding this research work through the project number (QU-IF-04-01-28436). The authors also thank Qassim University for technical support.

**Conflicts of Interest:** The authors declare no conflict of interest.

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
