# Peer review of "Leveraging Artificial Intelligence in Blockchain-Based E-Health for Safer Decision Making Framework"

_applsci, doi:10.3390/app13021035_

Round 1
Reviewer 1 Report
Authors proposed a model of integrating AI abilities with Blockchain. Few of my comments are as follows:
Scientific novelty of this paper is not very clear.
Please improve the grammar. Many sentences are hard to understand.
Provide the experimental setup of this paper. You ca provide code on github.
Practical implementation of this paper is not possible in my opinion. Please highlight your contribution.
Review references because some of them are unstandardized. The difference between your proposal and related works is not clear, you could to details better. I suggest add a comparative table in ''Related Literature'' to contrast your solution in front of related works. You can add few related papers such as: https://onlinelibrary.wiley.com/doi/full/10.1111/exsy.13173, https://ieeexplore.ieee.org/document/8861593
Author Response
Dear reviewer,
Thank you for your valuable comments certainly help to improve the quality of our work.
Please see our responses in the attached file.

Reviewer 2 Report
The manuscript presents a blockchain-based approach for e-health safer decision-making. Support vector machine and multi-layer perceptron classifiers are adopted as the algorithms. Overall, the manuscript is well-organized and easy to follow. However, we have the following comments.
* The e-heath task should be described in more detail. For example, what kind of classification task is considered, and what dataset is used?
* The role of blockchain in the machine learning models and how they are integrated should be discussed in more detail. For example, what nodes will run blockchain consensus, and what nodes will perform the machine learning model training and inference tasks?
* Some tables are presented in figures, which is not favorable. Also, some figures are not high-resolution enough.
* The related work section focuses too much on the preliminary knowledge while overlooking the related papers. Some important related works include "Blockchain-empowered Federated Learning: Challenges, Solutions, and Future Directions" and "Blockchain-based collaborative edge intelligence for trustworthy and real-time video surveillance".
* The algorithms can be supplemented with more explanations.
Author Response

(The authors gave the same response as above.)

Round 2
Reviewer 1 Report
Authors updated the paper and no further update requires from my side.
Author Response
Dear Reviewer,
Thank you for accepting the paper.

Reviewer 2 Report
Minor issue: the role of blockchain in work remains unclear.
Author Response
Dear reviewer,
Thank you for the comment:
To explain the role of blockchain, In the new version of the paper, we add a new subsection “1.3. Why blockchain?”
1.3. Why blockchain?
In our paper, the blockchain is used for two reasons:
- To protect the dataset against poisonous attacks. Since the used dataset is securely uploaded to the blockchain instead of publishing it in a shared repository. Indeed, the blockchain will guarantee the integrity of data.
- To protect machine learning techniques against evasion attacks. We perform this goal by embedding model decision functions as smart contracts in the Ethereum blockchain.
